# Energy-Efficient Random Number Generation Using Stochastic Magnetic Tunnel Junctions

**Nicolas Alder**
Hasso Plattner Institute
nicolas.alder@hpi.de

**Shivam Nitin Kajale**
MIT
skajale@mit.edu

**Milin Tunsiricharoengul**
MIT
milint@mit.edu

**Deblina Sarkar**
MIT
deblina@mit.edu

**Ralf Herbrich**
Hasso Plattner Institute
ralf.herbrich@hpi.de

## Abstract

(Pseudo)random sampling is a costly yet widely used method in machine learning. We introduce an energy-efficient algorithm for uniform Float16 sampling, utilizing a room-temperature stochastic magnetic tunnel junction device to generate truly random floating-point numbers. By avoiding expensive symbolic computation and mapping physical phenomena directly to the statistical properties of the floating-point format and uniform distribution, our approach achieves a higher level of energy efficiency than the state-of-the-art Mersenne-Twister algorithm by a minimum factor of 9721 and an improvement factor of 5649 compared to the more energy-efficient PCG algorithm. We provide measurements of the potential accumulated approximation errors, demonstrating the effectiveness of our method.

## 1 Introduction

The widespread implementation of artificial intelligence (AI) incurs significant energy use, financial costs, and $CO_2$ emissions. This not only increases the cost of products, but also presents obstacles in addressing climate change. At the heart of machine learning are sampling and random number generation. Examples include weight initialization, dropout regularization, or Markov chain Monte Carlo techniques.

Addressing these challenges, this paper proposes a novel hardware framework designed to enhance the energy efficiency of random number generation and sampling. We propose a novel uniform floating-point format sampling method utilizing stochastically switching magnetic tunnel junction (s-MTJ) devices as a foundation, achieving significant gains in both computational resources and energy consumption compared to current pseudorandom number generators. In contrast to existing generators, this device-focused strategy not only enhances sampling efficiency but also incorporates genuine randomness originating from the thermal noise in our devices. Simultaneously, this noise is crucial for the probabilistic functioning of the s-MTJs and is associated with low energy costs during operation.

Our contributions can be summarized as follows.

1. We present a novel, highly energy-efficient stochastically switching magnetic tunnel junction device which is designed to improve both the energy efficiency and precision of our sampling approach. The device is capable of generating samples from a Bernoulli distribution whose parameter $p$ can be controlled using a current bias.

38th Second Workshop on Machine Learning with New Compute Paradigms at NeurIPS 2024(MLNCP 2024).

2. We present a closed-form solution that defines the parameters for a collection of Bernoulli distributions applied to the bit positions of the floating-point format, leading to samples that adhere to a distribution without the need for symbolic calculations. Our simulations indicate that this hardware configuration surpasses existing random number generators in terms of energy efficiency by a factor of 5649 when using Float16. Additionally, our method achieves genuine randomness through the use of thermal noise in our hardware devices. In general, this approach is suitable for any entropy source device or even (pseudo)random number generator that can produce bits in a reliable (and efficient) Bernoulli fashion.

The structure of this paper begins by reviewing relevant work on random number generation in Section 2. Section 3 provides an introduction to the floating-point format, which is the format utilized to generate samples. In the Approach Section 4, we introduce the stochastically switching magneto-tunneling junction device being utilized in our approach. Following this, we outline a configuration for these devices to generate uniform floating-point samples, addressing the statistical challenge of mapping Bernoulli distributions to specific bitstring positions within the floating-point format. Section 5 illustrates the energy consumption of our approach and assesses potential approximation errors arising from the devices. The paper concludes with Section 6, where we summarize our findings and outline further research directions.

## 2 Related Work

A majority of artificial intelligence algorithms rely on random number generators. Random number generators (RNG) are employed for weight initialization or dropout in deep learning or taking random actions in reinforcement learning. In probabilistic machine learning, Markov-Chain-Monte-Carlo (MCMC) algorithms utilize them for sampling from proposal distributions or for making decisions on whether to accept or reject samples based on random draws.

Hence, the research community focused on the development of efficient random number generators [10] and their infrastructure [20, 15] shares similarities to this work. Physical (true) random number generators (TRNG) using physical devices is an active research field since the 1950s [11]. Currently used random number generators are often feasability-motivated free-running oscillators with randomness from electronic noise [19]. A very recent subfield are Quantum Based Random Number generators (QRNG) [12, 8, 19, 5]. The concept of employing stochastic magnetic tunnel junctions for random number generation has been investigated in recent years. Although these methods generally outperform traditional algorithmic random number generators in terms of energy efficiency, they lack the ability to directly produce results using the floating-point format [23, 3, 16, 17], which is critical for machine learning applications. Converting results to floating-point format later [4] introduces unnecessary overhead, reducing energy efficiency. In general, the unequal spacing characteristic of the floating-point format complicates the transition from integers, making it non-trivial to maintain all possible floating-point number candidates within a specific distribution. It should be noted that our conceptual approach can in principle be applied with any RNG that generates parametrizable Bernoulli distributions, given that they are sufficiently (energy-)efficient.

Antunes and Hill [1] accurately measured the energy usage of random number generators (Mersenne-Twister, PCG, and Philox) in programming languages and frameworks such as Python, C, Numpy, Tensorflow, and PyTorch, thus providing a quantification of energy consumption in tools relevant to AI. The energy measurements of this benchmark serve as baseline against our approach.

## 3 Preliminaries

We use the floating-point format as the number representation of interest as this is also the format that machine learning algorithms use. We define a generic floating-point number as follows:

$$x = \pm 2^{e-b} \cdot d_1.d_2 \ldots d_t, \tag{1}$$

where $e$ is the exponent adjusted by a bias $b$, $d_1.d_2 \ldots d_t$ represent the mantissa, $d_i \in \{0, 1\}$, and $d_1 = 1$ indicates an implicit leading bit for normalized numbers.

While our approach is generally applicable to any floating-point format, we demonstrate the approach for the Float16 format in this paper. The use of the Float16 format compared to formats with more

precision bits is advantageous in a real-world setting as it demands less rigor in setting the current bias for the s-MTJ devices, which is especially relevant for higher-order exponent bits.

In the following, we describe a Float16 number by its 16-bit organization

$$B = (b_0, b_1, \ldots, b_{15}), \tag{2}$$

where $b_{15}$ is the sign bit, $b_{14}$ to $b_{10}$ are the exponent bits with a bias of 15, and $b_9$ to $b_0$ are the mantissa bits. The implicit bit remains unexpressed. This arrangement represents the actual storage format of the bits in memory. By expressing the floating-point format in terms of its bit structure, we can directly map an s-MTJ device's output bit to its equivalent position in the Float16 format.

## 4  Approach

### 4.1  Probabilistic Spintronic Devices

Spintronic devices are a class of computing (logic and memory) devices that harness the spin of electrons (in addition to their charge) for computation [7]. This contrasts with traditional electronic devices which only use electron charges for computation. Spintronic devices are built using magnetic materials, as the magnetization (magnetic moment per unit volume) of a magnet is a macroscopic manifestation of its correlated electron spins. The prototypical spintronic device, called the magnetic tunnel junction (MTJ), is a three-layer device which can act both as a memory unit and a switch [14]. It consists of two ferromagnetic layers separated by a thin, insulating non-magnetic layer. When the magnetization of the two ferromagnetic layers is aligned parallel to each other, the MTJ exhibits a low resistance ($R_P$). Conversely, when the two magnetizations are aligned anti-parallel, the MTJ exhibits a high resistance ($R_{AP}$). By virtue of the two discrete resistance states, an MTJ can act as a memory bit as well as a switch. In practice, the MTJs are constructed such that one of the ferromagnetic layers stays fixed, while the other layer's magnetization can be easily toggled (free layer, FL). Thus, by toggling the FL, using a magnetic field or electric currents, the MTJ can be switched between its '0' and '1' state.

An MTJ can serve as a natural source of randomness upon aggressive scaling, i.e. when the FL of the MTJ is shrunk to such a small volume that it toggles randomly just due to thermal energy in the vicinity. It is worth noting that the s-MTJ can produce a Bernoulli distribution like probability density function (PDF), with $p = 0.5$, without any external stimulus, by virtue of only the ambient temperature. However, applying a bias current across the s-MTJ can allow tuning of the PDF through the spin transfer torque mechanism. As shown in Figure 5c-f of Appendix A, applying a positive bias current across the device makes the high resistance state more favorable, while applying a negative current has the opposite effect. In fact, by applying an appropriate bias current across the s-MTJ, using a simple current-mode digital to analog converter as shown in Figure 6a of Appendix A, we can achieve precise control over the Bernoulli parameter ($p$) exhibited by the s-MTJ. The $p$-value of the s-MTJ responds to the bias current through a sigmoidal dependence. A more detailed version of this section on the physical principles, device structure and simulations of the s-MTJ device can be found in Appendix A.

### 4.2  Random Number Sampling

This section describes the configuration of s-MTJ devices representing Bernoulli distributions for generating uniform random numbers in floating-point formats, particularly Float16. To apply this method to other floating-point formats, modify the number of total bits in Equation (3), (5) and (6) as well as the number of exponent bits in (8) and their positions in the format in variable $e$ of (6).

The configuration $C$ for a set of s-MTJ devices is defined as follows:

$$C = \{(b_i, p_i) \mid p_i \in [0, 1], b_i \in \{b_0, \ldots, b_{15}\}\}, \tag{3}$$

where each $p_i$ is the parameter of a Bernoulli distribution representing the probability of the corresponding Float16 format bit being '1' in the output.

The goal is to configure $C$ so that, with infinite resampling, the sequence $B_n$ of Float16 values converges to a uniform distribution $D$ over the full format. Formally, we seek $C$ such that:

$$\lim_{n \to \infty} P(B_n = b \mid C) = D(b), \text{ where } D = \text{Uniform}(-65504, 65504) \tag{4}$$

Table 1: Required 1-bit occurrences in a 3-bit exponent representation

|  | 1-Bit Count | | | | | | | |
| --- | --- | --- | --- | --- | --- | --- | --- | --- |
| $e_3$ | 0 | 0 | 0 | 0 | $2^4$ | $2^5$ | $2^6$ | $2^7$ |
| $e_2$ | 0 | 0 | $2^2$ | $2^3$ | 0 | 0 | $2^6$ | $2^7$ |
| $e_1$ | 0 | $2^1$ | 0 | $2^3$ | 0 | $2^5$ | 0 | $2^7$ |

In order to meet this condition, we need to assign each bit position $b_i$ of the Float16 format a probability $p_i$, representing the frequency of each bit's occurrence in a uniform Float16 distribution (Equations (5)-(8)). The mantissa bits are assigned a value of 0.5, as detailed in line (6), ensuring uniformity across the range they cover. This method extends to the sign bit, whose equal likelihood of toggling maintains the format's symmetry.

In floating-point formats, increasing the exponent doubles the range covered by the mantissa due to the base 2 system. Higher exponent ranges need more frequent sampling to maintain uniform coverage, as simply doubling sample occurrence from one range to the next does not preserve uniformity. Table 1 shows the number of 1-bits for each exponent in a 3-bit example. In general, one can see a specific overall pattern. Specifically, $e_1$ has four groups of size 1, $e_2$ has two groups of size 2, and $e_3$ has one group of size 1. More generally, the first count of any exponent group is always $2^{2^{i-1}}$. For the first exponent, groups are size 1 (excludable by $\mathbf{1}_{\{i>1\}}$). For other exponents, remaining 1-Bit counts in the first group are $\sum_{k=1}^{c-1} 2^{2^{i-1}+k}$, where $c = 2^{i-1}$ is the group size, depending on the position $i$ in the floating-point format. The count of groups based on bit position $i$ and total bits $e$ is $z = 2^{-i+e}$. The count sums for remaining groups are given by $\sum_{k=1}^{z-1} \sum_{g=1}^{c-1} 2^{2^{i-1}+2^i \cdot k+g}$, where $z$ is the number of groups and $c$ their size. The highest exponent bit $e_3$ with one group is excluded using $\mathbf{1}_{\{z>1\}}$. To find the probability of 1-Bit occurrences for each exponent $e_i$, divide by the total bits $2^{(2^e)} - 1$, which depends on the exponent bits $e$.

Combining everything, we derive the equation for the configuration $C$ as follows:

$$C = \{(b_i, p_i) \mid p_i \in [0,1], b_i \in \{b_0, \ldots, b_{15}\}\}, \text{ where} \tag{5}$$

$$p_i = \begin{cases} \frac{o_i-9}{2^{(2^e)}-1} & \text{if } i \in \{10, \ldots, 14\}, \\ 0.5 & \text{otherwise} \end{cases}, \text{ and} \tag{6}$$

$$o_i = 2^{2^{i-1}} + \sum_{k=1}^{c-1} 2^{2^{i-1}+k} \cdot \mathbf{1}_{\{i>1\}} + \sum_{k=1}^{z-1} 2^{2^{i-1}+2^i \cdot k} + \sum_{k=1}^{z-1}\sum_{g=1}^{c-1} 2^{2^{i-1}+2^i \cdot k+g} \cdot \mathbf{1}_{\{z>1\}}, \text{ and} \tag{7}$$

$$z = 2^{-i+e}, c = 2^{i-1}, e = 5. \tag{8}$$

After obtaining a sample $s$, min-max normalization can be applied to linearly transform it into a sample $s'$ that adheres to any specified uniform distribution within the Float16 range:

$$s' \sim \text{Uniform}(a,b) = a + \frac{(s + 65504) \cdot (b-a)}{131008}. \tag{9}$$

The transformation must be performed in a format exceeding Float16, like Float32 or a specialized circuit, to maintain numerical stability and precision, due to exceeding Float16 limits in the denominator of Equation (9). We assume special cases like NaNs or Infinities are discarded.

## 5 Evaluation

### 5.1 Energy Consumption of the s-MTJ Approach

Our work is motivated to use s-MTJ based random number generators for AI algorithms. We do not focus on advancing s-MTJ devices in the material sense. AI algorithms require an energy-saving random number generator that directly outputs uniformly in the floating-point format. We assess the energy consumption of our method against current (pseudo)random number generators employed in AI algorithms, incorporating a linear transformation (Equation (9) in section 4.2) to fit any specified

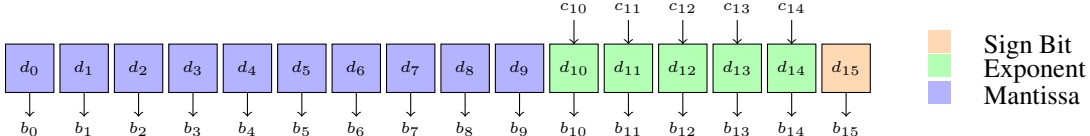

Figure 1: Hardware setup for sampling one value from a uniform Float16 distribution.

uniform number range. Although our technique inherently does not need this transformation, we include it in the energy-efficiency analysis to ensure a highly adaptable, fair, and conservative benchmark. This benchmark serves as comparison of our s-MTJ based sampling approach against traditional pseudorandom sampling approaches, not other s-MTJ devices in the material sense. The novelty and advantage of our approach against other s-MTJ based random number generators is the procedure of directly producing floating-point numbers without conversion from integer bitstreams. Any s-MTJ device implementation can be used with our approach.

Figure 1 depicts our hardware configuration for sampling a single Float16 value. Each $d_i$ is an s-MTJ device. The devices $d_{10}, \cdots, d_{14}$ for the exponent are equipped with 4 control bits to adjust the current bias $c_i$, which corresponds to the Bernoulli probability. The other devices are set to a fixed current bias equivalent to a Bernoulli of $0.5$. The resolution, which determines how accurately we can set the Bernoulli distributions for a device, is dependent on the number of control bits and is visualized in Figure 2. This Figure displays the specific Bernoulli values achievable with 4 control bits. Although additional control bits could allow for more precise settings, we restrict this number to 4 due to physical limitations in setting current biases in hardware with higher resolution while keeping the bias circuit simple (and hence energy-efficient). Our approach focuses on achieving high accuracy around a probability of 1 (cf. configuration in Section 5.2) by taking advantage of the characteristics of the sigmoid function, thus making 4 bits sufficient for achieving the required probability density function.

For our specific case, where the s-MTJs are being configured to generate a uniform distribution of Float16 samples, the $p$ for each s-MTJ is predetermined and fixed. All the mantissa bits and the sign bits require $p = 0.5$, which is exhibited by the s-MTJ without any current bias (cf. 4.1 and 4.2). Thus, these eleven s-MTJs do not require a current biasing circuit. The predetermined $p$-values for the five exponent bits correspond to specific current biases as shown in Figure 2, which amount to a total power consumption of $20.86\,\mu\text{W}$, as determined through SPICE simulations (details in Appendix C). For a sampling rate of $1\,\text{MHz}$, this corresponds to $20.86\,\text{pJ}$ biasing energy per Float16 sample. Additionally, reading the state of all the sixteen s-MTJs, assuming a nominal resistance of $1\,\text{k}\Omega$ and $10\,\text{ns}$ readout with $10\,\mu\text{A}$ probe current, amounts to a readout energy dissipation of $16\,\text{fJ}$ per Float16 sample.

Given a hardware accelerator-style architecture, our system is designed with an embarrassingly parallel structure, capable of producing samples every $1\,\mu\text{s}$. Energy-wise, there is no difference between parallel and sequential setups. Using min-max normalization, sampled intervals can be transformed efficiently into other intervals. It is reasonable that each of the five floating point operations mentioned in Equation 9 within a normalization circuit consumes about $150\,\text{fJ}$ on modern microprocessors [6], leading to an extra energy cost of $750\,\text{fJ}$ per sample.

Consequently, generating $2^{30}$ samples without linear transformation yields an energy consumption of

$$(16 \cdot 1\,\text{fJ} + 20.862\,\text{pJ}) \cdot 2^{30} = 22.42\,\text{mJ}. \tag{10}$$

Applying the transformation yields

$$(16 \cdot 1\,\text{fJ} + 20.862\,\text{pJ} + 750\,\text{fJ}) \cdot 2^{30} = 23.22\,\text{mJ}. \tag{11}$$

Our method's energy usage is compared to actual energy measurements taken by Antunes and Hill [1]. They benchmarked advanced pseudorandom number generators like Mersenne Twister, PCG, and Philox. This includes evaluations across original C versions (O2 and O3 suffixes refer to C flags) and adaptations in Python, NumPy, TensorFlow, and PyTorch, relevant platforms and languages for AI. Each measurement reports the total energy used to produce $2^{30}$ pseudorandom 32-bit integers or 64-bit doubles, which are common outputs from these generators. Often, specific algorithms and implementations are limited to producing only certain numeric formats (like integers or doubles),

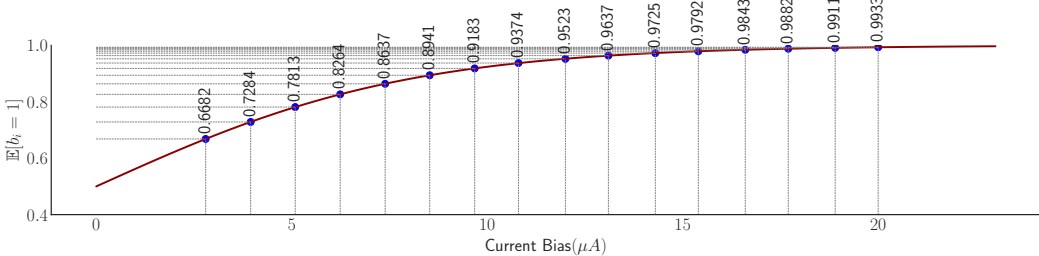

Figure 2: Possible Bernoulli resolutions for s-MTJ device with 4 control bits.

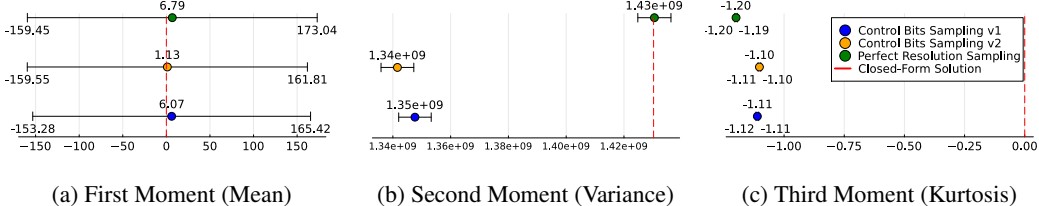

(a) First Moment (Mean)  (b) Second Moment (Variance)  (c) Third Moment (Kurtosis)

Figure 3: Physical approximation error comparison for the first three moments of the uniform distribution (s-MTJ-based approach vs. closed-form solution sampling). Second moment standard deviation omitted due to equivalence to the means.

particular bit sizes, or specific stochastic properties. As such, comparing different implementations and floating-point formats is somewhat limited. However, given that all implementations serve the same machine learning algorithms and that our energy consumption estimates show vast differences, this comparison is deemed both reasonable and significant.

Although our method introduces considerable energy costs due to transformations, the overall energy usage, when including linear transformations, is reduced by factor 5649 (pcg32integer) compared to the most efficient pseudorandom number generator currently available. Compared to the double-generating Mersenne-Twister (mt19937arO2), we obtain an improvement by factor 9721. We provide a full comparison against all benchmarked generators in Figure 7 of Appendix D.

## 5.2 Physical Approximation Error: Impact of Control Bits Resolution

The number of control bits in an s-MTJ device impacts both energy consumption and the precision of setting the energy bias, which in turn affects the available probabilities of obtaining bit samples. Figure 2 illustrates this relationship. This section evaluates the approximation error caused by imprecision in achieving a desired Bernoulli distribution.

Four control bits allow 16 distinct, uniformly spaced current biases for an s-MTJ device. The stability of reading a '1' or '0' from the device follows a sigmoid function, enhancing resolution near 0 and 1, but reducing it around 0.5. This effect is beneficial as it yields the configurations $c_{10}, c_{11}, \cdots, c_{14} = \{(10, 0.66666\overline{6}), (11, 0.80000), (12, 0.94118), (13, 0.99611), (14, 0.99998)\}$ for our hardware setup shown in Figure 1, as derived from Equations 5-8. Higher exponent bits demand greater precision than lower ones, highlighting the advantages of the Float16 format over larger formats due to the physical constraints in setting the energy bias.

To precisely analyze distribution shifts, we compared the first three moments (mean, variance, kurtosis) of the uniform Float16 distribution in Figures 3a, 3b, and 3c. We conducted $100\,000$ samples per measurement, repeating each measurement 100 times, and report the results as mean and standard deviations. We evaluated the empirical moments of these distributions against theoretical expectations using closed-form solutions. Control Bits Sampling v1 uses the closest distance, assigning equal probabilities of 0.9933 to $c_{13}$ and $c_{14}$. Control Bits Sampling v2 assigns probabilities of 0.9911 to $c_{13}$ and 0.9933 to $c_{14}$, testing whether having a difference is more effective than the closest distance method (see Figure 2). The mean values over all three moments are consistent for all

bit resolutions. Furthermore, the deviation in the second moment is relatively minor given its high absolute value in the closed-form expression.

Figure 9-12 of Appendix E visualizes samples using perfect resolution sampling and sampling that considers physical control bit boundaries. The distributions with approximation offsets show a slight bias, favoring values near zero (this is experimentally attributable to the offsets in exponent 4 and 5). However, this primarily accounts for only two bins in the overall range, each representing 0.25% of values. While the overall distribution remains unaffected, the effect can be removed by rejecting samples from the two bins in question, impacting approximately every 200th sample. These observations highlight that physical inaccuracies have minor effects. If necessary, these can be easily addressed through rejection from those bins, depending on the application's requirements. Although we assume that most applications will not be significantly affected, performance evaluations are required to verify this assumption (for any minor distribution shifts).

## 6   Conclusion and Future Work

We introduced a hardware-driven highly energy-efficient method for sampling uniform floating-point numbers, using stochastically switching magnetic tunnel junctions. This method includes a precise initialization for these devices and beats current state-of-the-art Mersenne-Twister by a factor of 5649.

We assessed the approximation error associated with the s-MTJ devices and our method. Findings show that the physical approximation error is negligible when sampling uniform random numbers. Future studies will explore the energy impact on specific algorithms in machine learning and validate the s-MTJ method by building a prototype including statistical randomness testing of the device [13, 11].

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

## A    Additional Information on the Spintronic Device

Spintronic devices are a class of computing (logic and memory) devices that harness the spin of electrons (in addition to their charge) for computation. This contrasts with traditional electronic devices which only use electron charges for computation. Spintronic devices are built using magnetic materials, as the magnetization (magnetic moment per unit volume) of a magnet is a macroscopic manifestation of its correlated electron spins. The prototypical spintronic device, called the magnetic tunnel junction (MTJ), is a three-layer device which can act both as a memory unit and a switch [7, 14]. It consists of two ferromagnetic layers separated by a thin, insulating non-magnetic layer. When the magnetization of the two ferromagnetic layers is aligned parallel to each other, the MTJ exhibits a low resistance ($R_P$). Conversely, when the two magnetizations are aligned anti-parallel, the MTJ exhibits a high resistance ($R_{AP}$). By virtue of the two discrete resistance states, an MTJ can act as a memory bit as well as a switch. In practice, the MTJs are constructed such that one of the ferromagnetic layers stays fixed, while the other layer's magnetization can be easily toggled (free layer, FL). Thus, by toggling the FL, using a magnetic field or electric currents, the MTJ can be switched between its '0' and '1' state.

An MTJ can serve as a natural source of randomness upon aggressive scaling, i.e. when the FL of the MTJ is shrunk to such a small volume that it toggles randomly just due to thermal energy in the vicinity. As schematically illustrated in Figure 4a, the self-energy of the magnetic layer is minimum and equal for the magnetization pointing vertically up or down, i.e. polar angle $\theta_M = 0^o$ or $180^o$, respectively. The self-energy is maximum for the horizontal orientation ($\theta_M = 90^o$). The corresponding energy barrier, $\Delta E$ dictates the time scale at which the magnet can toggle between the up and down oriented states owing to thermal energy. This time scale follows an Arrhenius law dependence [2], i.e.

$$\tau_{\uparrow\downarrow} = \tau_0 e^{\frac{\Delta E}{kT}}, \tag{12}$$

where, $\tau_0$ is the inverse of attempt frequency, typically of the order of 1 ns, $k$ is the Boltzmann constant and $T$ is the ambient temperature. The energy barrier for a magnet is $\Delta E = K_U V = \mu_0 H_K M_S V/2$, where $K_U$, $V$, $H_K$ and $M_S$ are the magnet's uniaxial anisotropy energy, volume, effective magnetic anisotropy field and saturation magnetization, respectively. $\mu_0$ is the magnetic permeability of free space. Thus, it can be observed that by reducing the volume $V$ of the magnetic free layer, we can make its $\Delta E$ comparable to $kT$ and achieve natural toggling frequencies of computational relevance, as shown in Figure 4b. Figure 5a shows a time-domain plot of the normalized state of such an s-MTJ, calculated using micromagnetic simulations with the MuMax3 package [21]. Further details on the micromagnetic simulations are included in Appendix B. A histogram of the resistance state of this s-MTJ is presented in Figure 5b. It is worth noting that the s-MTJ can produce such a Bernoulli distribution like probability density function (PDF), with $p = 0.5$, without any external stimulus, by virtue of only the ambient temperature. However, applying a bias current across the s-MTJ can allow tuning of the PDF through the spin transfer torque mechanism [18]. As shown in Figure 5c-f, applying a positive bias current across the device makes the high resistance state more favorable, while applying a negative current has the opposite effect. In fact, by applying an appropriate bias current across the s-MTJ, using a simple current-mode digital to analog converter as shown in Figure 6a, we can achieve precise control over the Bernoulli parameter ($p$) exhibited by the s-MTJ. Details on the current-biasing circuit are included in Appendix C. The $p$-value of the s-MTJ responds to the bias current through a sigmoidal dependence, as shown in Figure 6b.

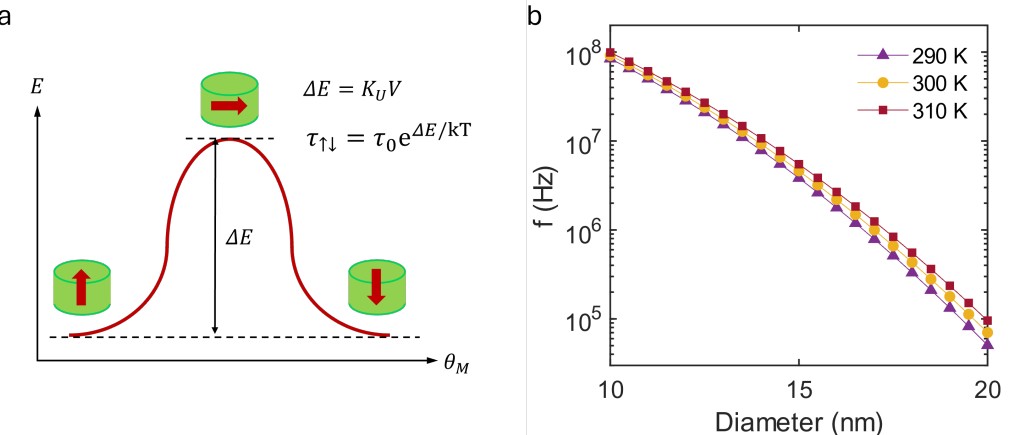

Figure 4: (a) Schematic illustration of the self-energy ($E$) of a nanomagnet with respect to the polar angle ($\theta_M$) of its magnetization (indicated by thick arrows). (b) Natural frequency of stochastic switching for a nanomagnet of a particular diameter at different temperatures.

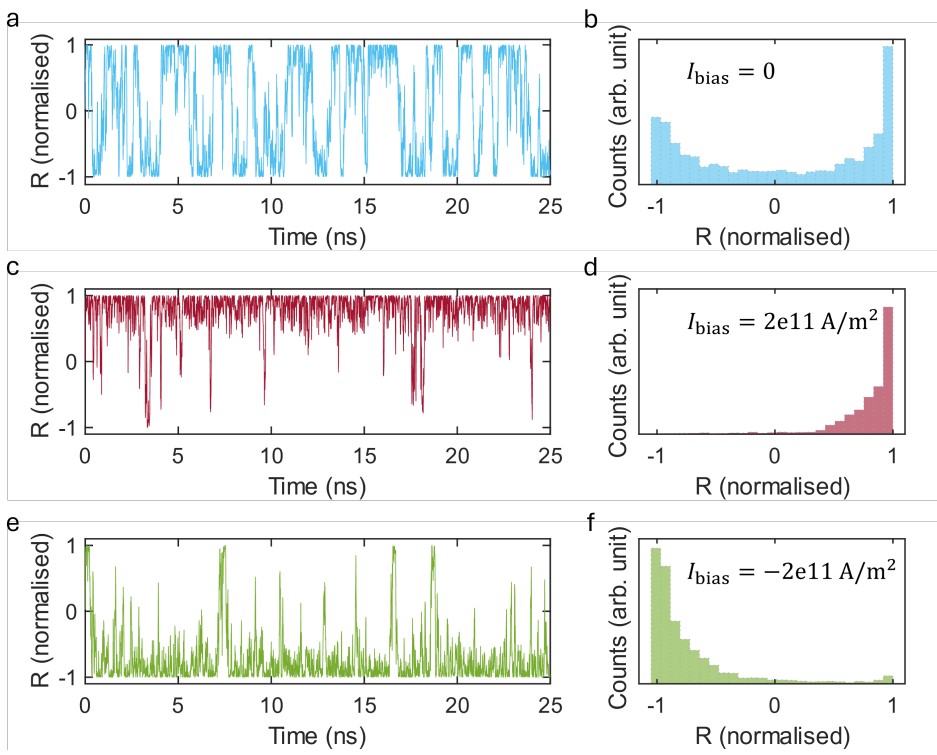

Figure 5: Dynamics of the normalized resistance of a stochastic MTJ for different bias current densities. (a) $I_{bias} = 0$ produces equal probability of observing the high or low state. (b) Histogram of the observed resistance state for $I_{bias} = 0$. (c, d) Trace and histogram of the observed resistance for a bias current of $2 \times 10^{11}$ A/m$^2$. (e, f) Trace and histogram of the observed resistance for a bias current of $-2 \times 10^{11}$ A/m$^2$.

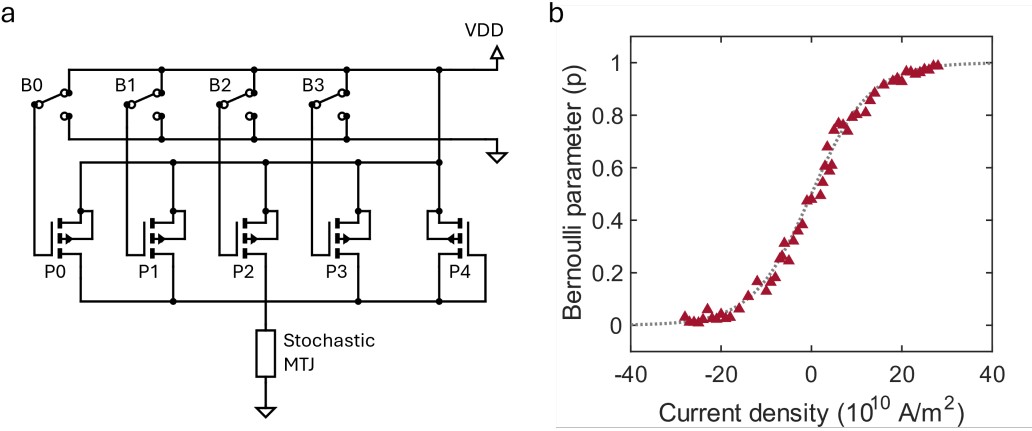

Figure 6: (a) Schematic diagram of a current-mode digital to analog converter for providing the biasing current to a stochastic MTJ. (b) Variation of the Bernoulli parameter of the stochastic MTJ with bias current. Red triangles are data point obtained from micromagnetic simulations, while the grey dotted line is a theoretical fit (sigmoid function).

## B  Micromagnetic Simulations

Dynamics of a ferromagnet's magnetization in response to external stimuli, like magnetic fields, currents or heat can be modelled using micromagnetic simulations. The magnetization dynamics can be described using a differential equation, known as the Landau-Lifshitz-Gilbert-Slonczewski (LLGS) equation:

$$\frac{d\vec{m}}{dt} = -\gamma \vec{m} \times \vec{H}_{\text{eff}} + \alpha \vec{m} \times \frac{d\vec{m}}{dt} + \tau_{\parallel} \frac{\vec{m} \times (\vec{x} \times \vec{m})}{|\vec{x} \times \vec{m}|} + \tau_{\perp} \frac{\vec{x} \times \vec{m}}{|\vec{x} \times \vec{m}|} \tag{13}$$

where, $\vec{m}$ is the normalized magnetization ($\vec{M}/|\vec{M}|$), and  are the gyromagnetic ratio and damping constant for the ferromagnet, x is a unit vector along the direction of applied electric current and, $\tau_{\parallel}$ and $\tau_{\perp}$ are current-induced torque magnitudes acting parallel and perpendicular to the current. $\vec{H}_{\text{eff}}$ is the effective magnetic field acting on the ferromagnet, which contains contributions from externally applied magnetic fields, exchange interactions, magneto-crystalline anisotropy, shape anisotropy, thermal fields, and demagnetization, among others.

The simulations results presented here are performed for a van der Waals (vdW) magnetic material, $Fe_3GaTe_2$ (FGaT) [22, 9]. Being a vdW material, FGaT has a layered structure which makes it an ideal candidate for building ultra-thin (monolayer) magnetic thin films of high quality needed for achieving stochasticity. FGaT also exhibits perpendicular magnetic anisotropy, which means its self-energy is lower for magnetization pointing out of plane as compared to the magnetization pointing in-plane. This property is crucial for building compact, nanoscale spintronic devices. The simulations are performed using the MuMax3 program [21], for devices shaped as circular discs. The values of different physical parameters used in the micromagnetic simulations are compiled in Table 2. Certain parameters, whose experimental values are not determined, are set to typical values for similar materials and are indicated as such. All simulations can be replicated using standard consumer-grade computers without requiring extensive resources.

Table 2: Parameters Used in Micromagnetic Simulations With the MuMax3 Code.

| Parameter | Value |
|---|---|
| Saturation magnetization ($M_S$) | $3.95 \times 10^4$ A/m [9] |
| Effective anisotropy field ($K_U$) | $3.02 \times 10^6$ A/m [9] |
| Permeability of free space ($\mu_0$) | $1.26 \times 10^{-6}$ kg·m/s$^2$·A$^2$ |
| Temperature ($T$) | 300 K |
| Gilbert damping constant ($\alpha$) | 0.02 (typical) |
| Exchange stiffness ($A_{\text{ex}}$) | $1.3 \times 10^{13}$ J/m |
| Thickness | 1 nm |
| Diameter | 2 nm |

## C   Power Estimation of the Current Biasing Circuit

The current biasing circuit was simulated using Cadence Virtuoso using the Global Foundries 22FDX (22 nm FDSOI) process design kit. The circuit has been designed for a maximum bias current of 20 $\mu$A to attain an s-MTJ with Bernoulli parameter $p = 0.99$. The current levels corresponding to $p = 0.67$ and $p = 0.99$ are divided into 4-bit resolution (Figure 2). The four bias bits (B0-B3) are fed to the transistors P0, P1, P2, P3 (LSB to MSB), which are sized to produce currents $I_0$, $2I_0$, $4I_0$ and $8I_0$, respectively, when the corresponding bias bit it '1'. A constant current $I_{\text{base}} = 2.82$ $\mu$A is additionally supplied through P4 to create a baseline of $p = 0.67$ for the s-MTJs. The transistors are operated at a low supply voltage of 0.35 V to achieve a small $I_0 = 1.14$ $\mu$A. Thus, each exponent bit can be set to its requisite Bernoulli parameter by appropriately setting the 4-bit bias word, and the power dissipation in the biasing circuit can be estimated for each of the exponent bits. Lengths of all the transistors are set to 20 nm. Width of P4 is set to 260 nm, while the widths of P0, P1, P2 and P3 are 100 nm, 200 nm, 400 nm, and 800 nm, respectively. As discussed in the main text, our proposed method requires only positive current biases for the stochastic MTJs. Thus, the unipolar current mode DAC proposed here suffices for our application. For more general use cases where both positive and negative bias currents may be needed, a bipolar current-steering DAC can be utilized.

# D  Energy Consumption of Random Number Generators

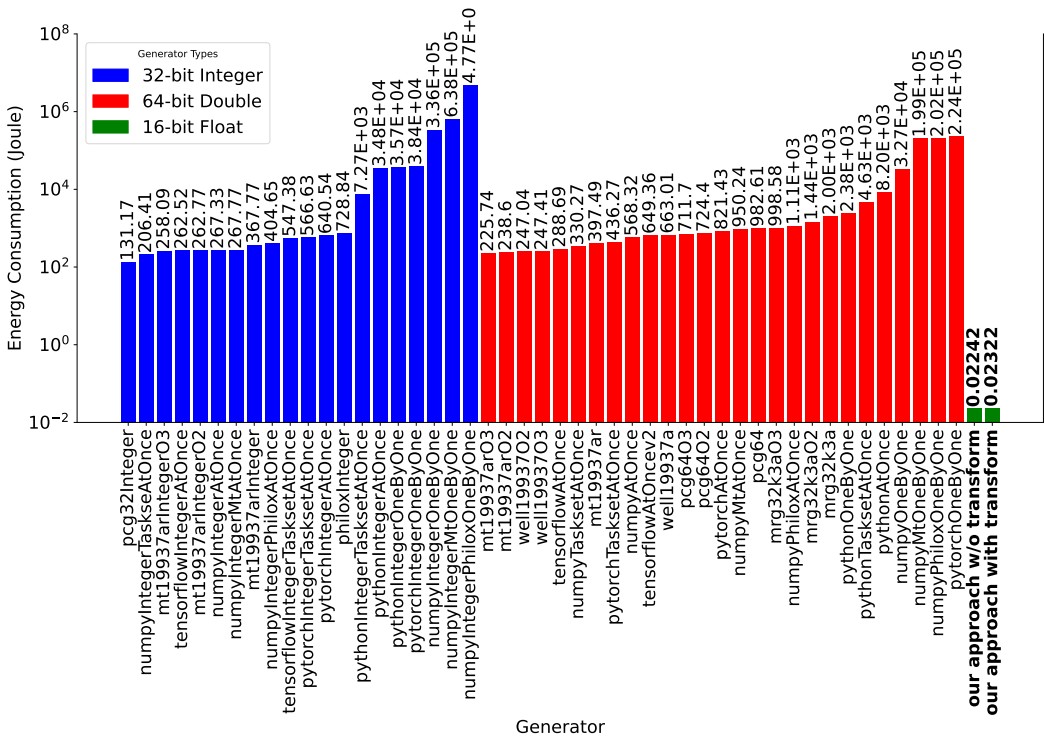

Figure 7: Power consumption analysis in Joules (logarithmic scale) for $2^{30}$ random numbers. Benchmarks were performed by Antunes and Hill [1].

# E   Additional Figures on Physical Approximation Error

Figure 8: Visualization of samples obtained with three different assumptions. Perfect Resolution Sampling assumes the precise values obtained from Equations 5-8 in Section 4.2. Control Bits Sampling v1 assumes the closest distance measure to actual obtainable control bits. Control Bits Sampling v2 assumes that each exponent bit should actually be different over closest distance, even if the physically closest distance would imply redundant values.

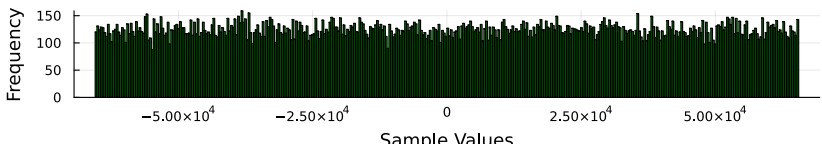

Figure 9: Histogram of $100\,000$ samples with $400$ bins over the full Float16 range obtained by Perfect Resolution Sampling.

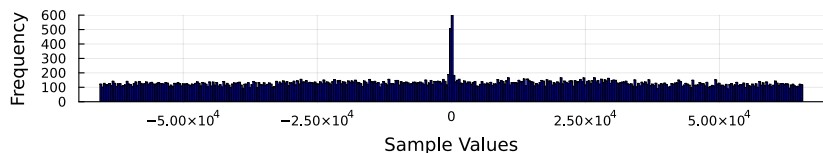

Figure 10: Histogram of $100\,000$ samples with $400$ bins spanning the full Float16 range obtained via Control Bits Sampling v1. The values show a slight bias, favoring those near zero. Each bin represents $0.25\%$ of the overall range. Flattening the distribution by rejecting samples from the two most overrepresented bins would affect only $0.5\ \%$ of samples.

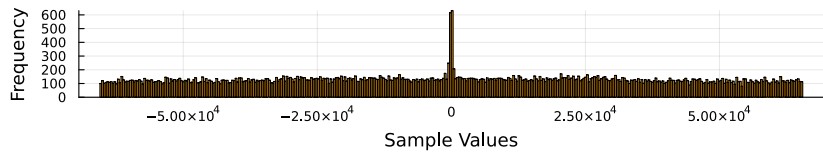

Figure 11: Histogram of $100\,000$ samples with $400$ bins spanning the full Float16 range obtained via Control Bits Sampling v2. The values show a slight bias, favoring those near zero. Each bin represents $0.25\%$ of the overall range. Flattening the distribution by rejecting samples from the two most overrepresented bins would affect only $0.5\%$ of samples.

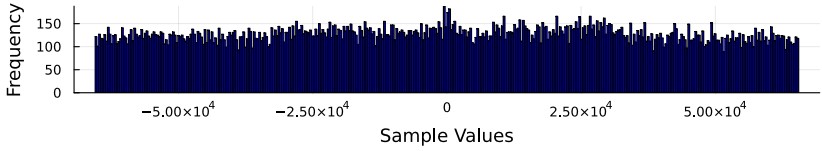

Figure 12: Histogram of $100\,000$ samples with $400$ bins spanning the full Float16 range obtained via Control Bits Sampling v1 with rejecting from the two most overrepresented bins around zero.

