# OpenReview forum: "Energy-Efficient Random Number Generation Using Stochastic Magnetic Tunnel Junctions"
_NeurIPS.cc/2024/Workshop/MLNCP — MLNCP Poster_

### Official Review · Reviewer_jcMn · 2024-09-29
**Promising work concerned with energy efficiency of random number generation**

**Rating:** 9
**Confidence:** 3

**Review:**

The paper addresses the problem of generating energy-efficient random numbers, with a focus on the applications in machine learning. This is an important area of research, as there is increasing concern about the sustainability and power consumption of machine learning algorithms.

The paper is very well written and clearly conveys the central message.

Strong points:

1. The paper develops a s-MTJ based floating point number generation scheme.
2. The authors provide extensive quantitative analysis of the energy and power consumption of their approach.

Weak points:

1. The authors approach generates uniform random numbers however in many machine learning applications Gaussian random numbers are used for weight initializations and other purposes. The paper could benefit from a paragraph that comments on how easy/difficult extend the present approach to Gaussian random numbers.

2. It would have been better to see these uniform numbers in a real-life application and see how the limitations of the proposed approach affect the performance of the algorithm.

Minor weak point (typo):
There is a missing Table reference on line 127.

---

### Official Review · Reviewer_ZGQv · 2024-10-02
**Energy-Efficient Random Number Generation with MTJ**

**Rating:** 5
**Confidence:** 3

**Review:**

The authors present a floating point RNG based on an MTJ.

The paper is well written with a few minor errors - for example a bad table reference on page 3.

I'm aware of several papers already proposing the use of current driven MTJ as RNGs. From a distance it is not exactly clear what the novelty of the paper is and should have been more clearly stated. Is it the way in which MTJ has been mapped to floating point format, the device, the circuit ?

Also energy efficiency is a major motivation of this work. But MTJs need to be driven with a large current (tens of uA here) and require current DACs for each random bit. Is it really better than a CPU or a dedicated ASIC for RNG ? Of course they are not T-RNGs, but it is not clear if a T-RNG is really vital in AI applications. A comparison between a more standard solution and your circuit would have been useful.

Due to (IMO) a lack of novelty I would say the paper is a boderline reject. If the paper had presented a fabricated circuit (or exp implementation) and not simulations it would better merit being accepted.

---

### Decision · Program_Chairs · 2024-10-10

Accept (Poster)